# RBM15 Promates the Proliferation, Migration and Invasion of Pancreatic Cancer Cell Lines

**DOI:** 10.3390/cancers15041084

**Published:** 2023-02-08

**Authors:** Hui Dong, Haidong Zhang, Xinyu Mao, Shiwei Liu, Wenjing Xu, Yewei Zhang

**Affiliations:** 1School of Medicine, Southeast University, Nanjing 210009, China; 2Hepatopancreatobiliary Center, The Second Affiliated Hospital of Nanjing Medical University, Nanjing 210011, China

**Keywords:** pancreatic cancer, RBM15, m6A, pancreatic cancer cell lines

## Abstract

**Simple Summary:**

This article reports that RBM15 is correlated with pancreatic cancer progression and is an essential prognostic biomarker in pancreatic cancer patients and is correlated with progression by bioinformatic integrated analysis, clinical information statistical analysis and wet experiments.

**Abstract:**

(1) Background: Pancreatic cancer is increasingly becoming the leading cause of cancer deaths worldwide. In eukaryotic cells, m6A is the most abundant mRNA methylation modification. (2) Methods: We performed a bioinformatic multidimensional analysis using files containing the clinical data of patients and m6A-related gene expression differences downloaded from web-based databases, and performed a statistical analysis and image mapping mainly using R-package. Next, we studied the RBM15 expression in cancer and paracancerous tissues. We then validated these findings in two cell lines by western blot, PCR, Transwell, CCK-8, and EDU animal models. (3) Results: We discovered that RBM15 was highly expressed in pancreatic cancer patients and that it is a significant cause of poor prognosis. Its association with lymphatic T cell family aggregation was established through immune infiltration analysis. A retrospective analysis of data from clinical patient specimens revealed that high expression of RBM15 in patients was closely and positively correlated with preoperative glucose values, gender, and lymphocyte counts. Results from cellular experiments and animal experiments indicated that when the RBM15 gene was silenced, cell proliferation, migration, and metastasis were inhibited. (4) Conclusions: We propose that RBM15 plays a key role in the progression of pancreatic cancer by promoting tumor proliferation, migration and metastasis.

## 1. Introduction

According to data from the National Cancer Center, pancreatic cancer is the tenth most common form of malignant tumor in China. Furthermore, it ranks sixth and seventh in the causes of tumor-related deaths in men and women, respectively, and the 5-year survival only ranges from 7.2% to 9% [1]. In the United States, the 5-year survival rate is about 10%, and pancreatic cancer is becoming an increasingly common cause of cancer mortality [2]. In eukaryotic cells, m6A is the most abundant mRNA methylation modification. It affects every process of the RNA life cycle [3], such as mRNA stability, splicing, and precursor miRNA processing. In addition, m6A participates in biological processes such as tissue development, stem cell repair and differentiation, DNA damage response, biorhythm regulation, and natural immune regulation [4,5]. In intracellular homeostasis, m6A is maintained by the methyltransferase complex and demethylase. It is messaged by m6A-specific recognition proteins, thereby building an efficient and ordered m6A regulatory network. Three major functional enzymes are involved in m6A modification, including adenosine methyltransferase “writers” (METTL3, METTL14, METTL16, WTAP, KIAA1429, RBM15, and ZC3H13), demethylase “erasers” (FTO and ALKBH5) and methylation “reader” proteins (YTHDF1/2/3, YTHDC1, HNRNPA2B1, and HNRNPC) [6]. Research has established that m6A modifications are closely associated with many forms of cancer development, such as the promotion of stem cell self-renewal in leukemia [7], the stimulation of stem-like cell differentiation in glioblastoma [8], the promotion, proliferation, and invasion of hepatocellular carcinoma cell lines [9], disease progression in breast cancer [10], etc. However, the role of RNA binding motif protein 15 (RBM15) as a “writer” in the process of m6A modification and how it affects the progression of tumors is not yet known.

RBM15 is an evolutionarily conserved spen family protein involved in cell fate determination. It is also an RNA binding protein encoded by the *RBM15*/*OTT* gene located on chromosome 1 [11]. RBM15 binds to pre-mRNA-specific sites in introns, recruits SF3B1 to splice regions, and promotes alternative RNA splicing at different cellular stages [12]. As observed by confocal microscopy, RBM15 is predominantly located within nuclear RNA splice sites, suggesting that RBM15 is involved in RNA splicing [13]. In addition, RBM15 plays a key role in the physiopathological processes of organisms [14,15], and abnormal mutations of RBM15 have been widely observed in several tumor studies, including adrenocortical carcinoma [16], acute myeloid leukemia [17], lung adenocarcinoma [18], squamous cell carcinoma of the larynx [19], and interstitial thyroid cancer [20].

Given this knowledge, we performed a bioinformatic multidimensional analysis using files containing the clinical data of patients and m6A-related gene expression differences downloaded from web-based databases, and performed a statistical analysis and image mapping mainly using the R-package. In addition, through a series of in vitro and in vivo experiments, we basically determined the oncogenic role of the high expression of RBM15 in pancreatic cancer progression. We hypothesize that RBM15 will play an important role in the diagnosis and treatment of pancreatic cancer in the future.

## 2. Materials and Methods

### 2.1. The Dataset Sourcing and Pre-Processing of Healthy and Pancreatic Cancer Patients

We downloaded the gene expression files of healthy controls and pancreatic cancer samples from The Cancer Genome Atlas (TCGA) database and the Genotype-Tissue Expression (GTEx) library via the UCSC Xena website. In addition, we searched for public gene expression data and complete clinical data from the Gene Expression Omnibus (GEO) and TCGA databases. The data from the GEO database is encoded as GSE21501. After downloading the gene expression data from the UCSC Xena website and the GEO database, we performed pre-processing tasks such as conversion ID, FPKM value conversion, and data merging. Next, we collated and merged clinical data for subsequent analysis. The copy number files of m6A-related genes in pancreatic cancer samples were downloaded from the UCSC Xena website. To further evaluate the role of m6A-related genes in the Kyoto Encyclopedia of Genes and Genomes (KEGG) pathway, we downloaded the gene set files of the KEGG pathway from the Gene Set Variation Analysis (GSVA) network.

### 2.2. Construction of m6A Gene Expression Differential Analysis

We identified 23 m6A-related genes, which we called regulators, through a literature search. Among them were eight writers: METTL3, METTL14, METTL16, WTAP, VIRMA, ZC3H13, RBM15, and RBM15B; 13 “readers”: YTHDC1, YTHDC2, YTHDF1, YTHDF2, YTHDF3, HNRNPC, FMR1, LRPPRC, HNRNPA2B1, IGFBP1, IGFBP2, IGFBP3, and RBMX; and two erasers: FTO and ALKBH5. In addition, we obtained 178 pancreatic cancer samples and 171 normal pancreatic tissue samples from the UCSC Xena website. RNA sequencing data were used to further analyze the differences in m6A gene expression in normal and cancerous tissue. We downloaded the copy number files of m6A-related genes in pancreatic cancer samples from TCGA-PAAD at the UCSC Xena website for copy number variation (CNV) analysis. In addition, the gene expression files of pancreatic cancer samples obtained from the TCGA and GEO databases were used to analyze the mutation frequency of m6A-related genes in pancreatic cancer, pancreatic cancer tumor mutation load, and the expression differences of related genes.

### 2.3. Construction of m6A Typing, Sample Gene Expression Differences, and Immune Cell Infiltration Enrichment Analysis

Based on their expression in the samples, we clustered and typed the m6A-related genes. After obtaining the m6A typings, we determined the variations among the different m6A typings according to gene expression. From a clinical perspective, the relationship between different m6A typings and patient prognosis was further analyzed. By applying the gene set files of the KEGG pathway from the GSVA network, we studied differences in the expression of function and pathway between various m6A typologies and then used GO enrichment to analyze the involvement of various genes in cellular physiological functions. A single-sample gene set enrichment analysis (ssGSEA) on the relationship between immune genes and different m6A typings was conducted to determine the relative abundance of each TME-infiltrating cell in the samples. Furthermore, a principal component analysis (PCA) was carried out to score the relevant genes and cluster the different samples according to m6A typing.

### 2.4. Construction of Correlation Analysis between m6A-Related Genes and Patients’ Clinical Data

The complete clinical data of 288 pancreatic cancer patients obtained from the GEO and TCGA databases included patient age, gender, survival status, survival time, and tumor grade. The data downloaded from the website were processed by converting IDs, merging values, etc., for subsequent analysis. We analyzed the relationship between the high and low expression of m6A-related genes in pancreatic cancer patients and the survival time of patients after surgery. We then plotted the prognostic network of m6A-related genes to establish which genes were closely associated with patient prognosis. By clustering the gene expression, we typed these genes to obtain m6A typings and constructed a survival analysis of their association with prognosis. After obtaining the scores of the m6A prognosis-related genes from the PCA analysis, the scores were statistically analyzed to determine the optimal cut-off values. Next, the genes were categorized into high and low groups. After obtaining the m6A scores, we performed the following analyses: the difference between m6A scores and patient prognosis according to the different survival times of the samples; the relationship between high and low scores and immune cell infiltration; the association between high and low scores, m6A typing, and prognosis-related genotyping; the correlation between scores, genotyping, and tumor mutation load; and the relationship between high and low m6A scores and patient prognosis under different clinicopathological grading survival analyses. Finally, to determine a relationship between tumor mutation load and patient prognosis, we distinguished between high and low tumor mutation load groups.

### 2.5. Experimental Testing

Three cell lines: HPNE, CFPAC-1, and BxPC-3, purchased from Procell Life Science & Technology, were used for the cellular experiments. The expression of the *RBM15* gene in these three cell lines was detected by qPCR followed by immunohistochemical staining to reveal RBM15 protein expression in pancreatic cancer tissue and paracancerous normal tissue. The PCR reagent were purchased from Servicebio. Additionally, cell transfection was applied to knock down the expression of the RBM15 gene. The CCK-8 assay, scratch assay, Transwell assay, and EdU assay were then carried out. The Si-RNA and Sh-RNA were purchased from Genechem, and the nude mice were purchased from Charles River. The cells were treated well in advance and divided into knockdown and control groups. An amount of 100 μL (including 10^6^ cells) of cell suspension was taken and injected into the subcutaneous area of the left hind limb of nude *mice*. The needle was inserted 1.5 cm and pushed slowly. After injection, we observed the state of nude *mice* and made daily records. This clarified the role of the *RBM15* gene in pancreatic cancer cell lines.

### 2.6. Statistical Analysis and Image Plotting

We used R (version 4.0.5) for statistical analysis and plotting. In the data processing stage, we used the limma package, sva package, and the reshape2 package to transform the values and merge the data. By using the Survminer R package and applying the “surv-cutpoint” function, we categorized the m6A scores by applying the optimal cutoff value. To reduce the batch effect of the calculations, we then divided the patients into high and low m6A score groups according to the maximum selected log-rank statistic. The specificity and sensitivity of the m6A scores were assessed using the receiver operating characteristic (ROC) curve, and the area under the curve (AUC) was quantified using the pROC package. We used the Kaplan-Meier method to generate survival curves for prognostic analysis under various types of risk factors and employed the log-rank test to determine the significance of the differences. Next, we identified independent prognostic factors by multivariate Cox regression models in which the information of the enrolled patients was detailed and complete. We used the RCircos package to map the variation of genes on chromosomes, the ggpubr package to visualize the differences in gene expression between normal and pancreatic cancer tissue, and the maftools package to illustrate the degree of variation for m6A-related genes in pancreatic cancer. Additionally, we used ColorBrewer, palettes, and other packages for coloring and other modifications.

## 3. Results

### 3.1. Expression of m6A Gene in Pancreatic Cancer

We compared the RNA sequencing data from 178 pancreatic cancer samples and 171 normal pancreatic tissue samples to determine the differences in m6A gene expression. This study included 23 m6A regulators, including eight “writers”, two “erasers”, and 13 “readers”. The genes *RBM15*, *RBM15B*, *ZC3H13*, *YTHDF1*, *YTHDF2*, *YTHDF3*, *IGFBP1*, *IGFBP3*, and *ALKBH5* were all found to be more highly expressed in pancreatic cancer tissue than in normal tissue. Of these regulators, there were writers as well as readers regulating m6A, but no erasers. Furthermore, genes such as *METTL3*, *METTL14*, *METTL16*, *WTAP*, *ZC3H13*, *YTHDC1*, *YTHDC2*, *HNRNPC*, *FMR1*, *LRPPRC*, *HNRNPA2B1*, *IGFBP2*, *RBMX*, *VIRMA*, and *FTO* were expressed at higher levels in normal tissue (Figure 1A). Figure 1B shows the location of the m6A regulators on the chromosome for CNV alterations.

### 3.2. m6A Gene and Patient Prognosis in Pancreatic Cancer

Based on the clinicopathological data of 288 pancreatic cancer patients, we further analyzed the relationship between the 23 regulators and the clinical data. The expression of *RBM15*, *RBM15B*, *ZC3H13*, *YTHDF1*, *YTHDF2*, *YTHDF3*, *IGFBP1*, *IGFBP3*, and *ALKBH5* was significantly higher in pancreatic cancer tissue than in normal tissue. Furthermore, the expression of *RBM15*, *ZC3H13*, *YTHDF2*, *YTHDF3*, and *IGFBP2* was negatively correlated with patient prognosis. Thus, higher gene expression resulted in a shorter postoperative survival time. In addition, the expression of *FTO*, *FMR1*, *HNRNPA2B1*, *HNRNPC*, and *VIRMA* was positively correlated with patient prognosis (Figure 2A). An analysis of the constructed prognostic network diagram indicated that genes such as *HNRNPC*, *RBM15*, and *VIRMA* were risk factors strongly associated with prognosis (Figure 2B).

### 3.3. Patterns of m6A Methylation Modifications

Based on the expression of the 23 m6A methylation regulators in pancreatic cancer patients, we performed an unsupervised cluster analysis using the R function “ConsensusClusterPlus”, which is explained on the Bioconductor website as an “algorithm for determining cluster count and membership by stability evidence in unsupervised analysis”. The genes clustered together had similar expression patterns in pancreatic cancer samples, which may indicate the co-regulation of these genes. In addition, genes from the same cluster may perform similar cellular functions, which helps annotate newly discovered genes. When the optimal k value is 3, the clustering is the most ideal, which is defined as the classification A, B, and C. We obtained three different m6A typings: type A in 151 cases, type B in 61 cases, and type C in 76 cases (Figure 3A). In addition, we determined that there was a relationship between m6A typing and patient prognosis. Results from the heat map (Figure 3B), made by correlating patients’ clinicopathological features and gene expression showed that *METTL16*, *WTAP*, *VIRMA*, *ZC3H13*, *RBM15*, *RBM15B*, *YTHDC1*, *YTHDC2*, *YTHDF1*, *YTHDF2*, *YTHDF3*, *HNRNPC*, *FMR1*, *LRPPRC HNRNPA2B1*, *RBMX*, *FTO*, and *ALKBH5* exhibited a higher expression of m6A fraction B. The pathological grading stages of patients in this aggregated segment were mostly N1-N3 and T3-T4, with more advanced disease staging. However, *IGFBP2* showed a higher expression of genes associated with m6A fraction C in pancreatic cancer prognosis. We found that the taxon with typing B had the worst prognosis, while the modified pattern of m6A typing C had a clear survival advantage (Figure 3C). Subsequently, we compared the gene expression of m6A-related genes in biological life information pathways between typing C and B (Figure 3D). We concluded that m6A typing C plays an important role in the basal metabolic activities of the body, including glycine, serine, and threonine metabolism, diabetes in young adults, proximal tubular bicarbonate recycling, cardiac muscle contraction, arginine and proline metabolism, arachidonic acid metabolism, linoleic acid metabolism, retinol metabolism, and cytochrome P450 metabolism of xenobiotics. Surprisingly, m6A fraction B has a vital role in the immune processes of the body and tumor progression, such as the T cell receptor signaling pathway, basic transcription factors, ubiquitin-mediated protein hydrolysis, renal cell carcinoma, colorectal cancer, chronic granulocytic leukemia, and neurotrophin signaling pathways. This suggests that m6A typing B is more actively involved in human pathophysiological processes and plays a crucial part in T cell-based cellular immunity. Additionally, we further studied the immune cell infiltration of each m6A typing by ssGSEA (Figure 3E). Here, CD4+ T cells, natural killer T cells, regulatory T cells, type 1 T helper cells, type 17 T helper cells, type 2 T helper cells, gamma delta T cells, activated dendritic cells, immature dendritic cells, macrophages, and natural killer cells were particularly active in each m6A typing involved in pancreatic cancer disease progression. Among them, activated dendritic cells aggregated more significantly in m6A type A, while the remaining cells showed a higher tendency to aggregate in m6A typing B. This echoes our previous results, which suggested that m6A typing B was involved in pancreatic cancer disease progression through the T cell family. However, whether it promotes or suppresses tumors remains to be established.

### 3.4. Clinical and Transcriptomic Features of m6A-Related Phenotypes

We identified the differential genes as well as the intersecting genes between the phenotypes by comparing the expression of genes in their corresponding samples under different m6A phenotypes. The Wayne diagram (Figure 4A) indicates that there were 314 intersecting genes between the three subtypes of m6A typings. By implementing GO enrichment analysis (Figure 4B), we observed the role played by the intersecting genes in biological processes, cellular localization, and molecular functions. KEGG pathway analysis (Figure 4C) revealed that the intersecting genes were mainly associated with the T cell receptor signaling pathway, Yersinia pestis infection, NOD-like receptor signaling pathway, and ubiquitin-mediated protein hydrolysis. To further identify the prognosis-related genes in the intersecting genes, we conducted an unsupervised cluster analysis of the samples based on gene expression to classify the prognosis-related genes into gene typings A-C (Figure 4D). Our data analysis (Figure 4E) showed that m6A regulator gene expression was considerably different in all genotyped samples. The expression was generally higher in genotype B, which also indicated that m6A modifications existed in each genotype subtype. A survival analysis (Figure 4F) revealed that samples with genotype C had a greater survival advantage, while those with genotype B had the worst prognosis. We also found that genotype B resulted in a poorer prognosis and that such patients generally had a posterior tumor grade (Figure 4G).

### 3.5. m6A Score and Clinicopathological Characteristics of Pancreatic Cancer Patients

Considering the individual heterogeneity and complexity of the samples and based on the expression of prognosis-related genes, we constructed a scoring system to quantify the m6A modification pattern in individual pancreatic cancer patients, titled “m6Ascore”. The m6A score was calibrated according to the optimal cut-off value of 7.794986, which was established using the R package Survminer. The scores were then divided into two groups of high and low scores. Alluvial plots were used to visualize changes in the attributes of individual patients (Figure 5B). By performing an immune correlation analysis (Figure 5C), we observed that the m6A score correlated with immune cells such as CD4+ T cells, regulatory T cells, type 1 T helper cells, gamma delta T cells, natural killer T cells, activated dendritic cells, CD56+ natural killer cells, and natural killer cells, and the difference was significant. By statistically typing the m6A score for each m6A typing as well as the samples under gene typing, we found that the B-type m6A score was substantially higher than the other two types, regardless of the typing (Figure 5D,E). Additionally, we statistically analyzed the survival status of patients in different m6A scoring groups (Figure 5A,H) and found that patients in the high scoring group had higher mortality rates than those in the low scoring group. We further investigated the tumor mutation load of pancreatic cancer (Figure 5F) and learned that the prognosis of patients with a high tumor mutation burden (TMB) was worse than patients with a low TMB. Next, we combined the high and low m6A score with the high and low TMB, respectively, and compared the interference between the four different combinations on the survivability of pancreatic cancer patients. We found that, regardless of the performance of the tmb group, as long as the score of m6a is high, the prognosis is poor (Figure 5G). Investigations into the relationship between the prognosis of pancreatic cancer patients with different m6A scores and their disease stage (Figure 5I) revealed that the prognosis of patients with disease stages between T3 and T4 correlated to their m6A scores. The prognosis of patients with high m6A scores was considerably worse, while the prognosis of patients in the lower disease stages did not significantly correlate with the m6A score. We then compared PD-L1 gene expression with m6A score in pancreatic cancer tumor samples (Figure 5J). We established that PD-L1 expression was higher in the high-scoring group and lower in the low-scoring group, with a statistically significant difference. This also suggests that the high-scoring group may be more suitable for tumor immunotherapy.

From Figure 3B we can see that *RBM15* is located in m6a cluster B, while Figure 4E shows that *RBM15* is also highly expressed in geneCluster B. Figure 5B indicates that clusters A and C have low scores (lower than 7.794986), while typing B has higher scores. Of course, we do not yet fully know the detailed genes of this part. However, a survival analysis of the different types revealed that the RBM15 gene is closely related to high scores and, consequently, higher mortality.

### 3.6. Clinical Significance of High Expression of RBM15 in Pancreatic Cancer Patients

Based on bioinformatics technology, we established that RBM15 played a vital role in the progression of pancreatic cancer. Its expression in pancreatic cancer tissue was significantly higher than in normal tissue, and the difference was statistically significant. Furthermore, it was negatively correlated with patient prognosis, as higher gene expression led to shorter postoperative survival times. An analysis of the constructed prognostic network diagram indicated that RBM15 was a risk factor closely related to prognosis. We carried out further cellular experiments by qPCR (Figure 6A), which indicated that the RBM15 gene was highly expressed in pancreatic cancer cell lines (CFPAC-1, BxPC-3), with a significant difference. Through a western blot experiment, we found that RBM15 protein does show a consistent high expression trend in the above-mentioned pancreatic cancer cell lines (Figure 6B). Moreover, by conducting immunohistochemical experiments (Figure 6C), we discovered that RBM15 was considerably more highly expressed in pancreatic cancer tissue than paracancerous normal tissue. Figure 6D presents the distribution of immunohistochemical scores and indicates that the scores of cancerous tissue were significantly higher than those of paracancerous tissue. We found that concerning the immunohistochemical score, the expression of RBM15 was higher in pancreatic cancer patients. From the Receiver operating characteristic curve (ROC curve), we determined the cut-off value of the score, which was 5 (Figure 6D). Figure 6E presents the distribution of immunohistochemical scores (*p* < 0.001). Table 1 displays the association between the RBM15 score and clinicopathological features. We divided the patients into high and low groups based on the critical values and found that the high score group was correlated with patient gender, preoperative venous glucose value, and lymphocyte infiltration (*p* < 0.05). However, other clinical characteristics, such as age, tumor size, and neutrophil ratio, did not present a statistically significant association (*p* > 0.05).

We further observed the effect of RBM15 on pancreatic cancer cells by knocking down its expression in pancreatic cancer cell lines. We discovered that cell migration, metastasis, and proliferation were affected by the knockdown of *RBM15* in two cell lines, CFPAC-1 and BxPC-3. The small interfering RNAs were used to construct pancreatic cancer cell lines with low *RBM15* expression. The qPCR results in Figure 7A,B show that all three small interfering RNAs effectively knocked down the expression of *RBM15* in pancreatic cancer cell lines CFPAC-1 and BxPC-3. Figure 7C,D shows the results of the CCK-8 experiment in the CFPAC-1 and BxPC-3 cell lines after knocking down the *RBM15* gene. It shows that the proliferation or viability of the cells fell significantly after silencing *RBM15*. Figure 7E,F shows the results of the scratch assay in the CFPAC-1 and BxPC-3 cell lines. It indicates that cell migration was inhibited after RBM15 was knocked down. Figure 7G,H, which display the results of the Transwell experiments in cell lines CFPAC-1 and BxPC-3, show that cell migration was inhibited after RBM15 was knocked down. Also, Figure 8A illustrates the results of the BxPC-3 cell line for the EdU uptake assay and reveals that cell migration was inhibited after RBM15 knockdown. Figure 8B shows that in the nude mouse subcutaneous tumorigenesis model, the knockout *RBM15* group had smaller tumor volumes than the non-knockout group.

## 4. Discussion

Through its regulators, the m6A modification plays a significant role in the process of organismal inflammation, tumor immunity, and tumor progression. However, the study of m6A modification patterns in tumor microenvironment immune cell infiltration during tumor progression has not yet been fully explored. Therefore, to provide further guidance in tumor immunotherapy, we conducted a multifaceted analysis to evaluate a total of 23 regulators for three types of m6A modification. We first performed a statistical analysis of the gene expression of these 23 regulators based on database gene expression data. We discovered that most of the regulators with m6A modification patterns have relatively high gene expression, and this elevation is also associated with poor patient prognosis. Among these regulators, the “writer” *RBM15* gene is the most typical representative. These m6A modification pattern-related genes are highly expressed in pancreatic cancer disease progression, which in turn contributes to disease development. The three m6A types have different survival rates, and the poor prognosis of type B patients is mainly due to the high expression of the m6A “regulator” genes and the advanced disease stage compared to type C patients. Additionally, each type has its distinctive features. Type B is characterized by T cell-based immune activity, type C is depicted by basal metabolic activity, while type A is not clearly defined. This suggests that the m6A modification pattern is involved in the disease progression of advanced pancreatic cancer by activating the T cell family to participate in tumor migration, invasion, and metastasis. The enrichment of CD4+ T cells, regulatory T cells, type 1 T helper cells, gamma T cells, and natural killer T cells is a distinctive feature. Li et al. demonstrated that the deletion of the m6A “writer” protein METTL3 in mouse T cells disrupted T cell homeostasis and differentiation, thereby reducing the incidence of colitis [21]. Furthermore, Yao et al. reported that the conditional deletion of METTL3 in CD4+ T cells impaired the intracellular TFH differentiation and germinal centers in mice, which in turn regulated TFH cell production [22]. This suggests that m6A-related genes are involved in the pathological process of tumors through the T cell receptor signaling pathway in pancreatic cancer progression.

In one study, D J McGrail et al. [23]. found that a high tumor mutation burden fails to predict immune checkpoint blockade response across all cancer types, and in our bioinformatics study, high TMB predicted poor survival, but when analyzed in conjunction with m6A scores, we found that the effect was weakened. Addeo et al. pointed out that TMB is a challenging biomarker with potential use in clinical practice that is still unclear [24]. Because the mechanism is not clear, many treatment options become particularly important. Some clinical studies in other cancers have shown that patients with high TMBs do not necessarily respond better to PD-L1. Meiri E et al. found that smaller studies of patients with MSS metastatic colorectal cancer have shown mixed results for TMB as a predictive biomarker [25]. John H. Strickler et al. summarized that mutation quality is more important than mutation quantity [26]. Therefore, combined with our findings, high tmb has a poor prognosis, which may be the reason that its mutation quality does not cause sufficient recognition of immune cells, and thus the immune microenvironment of the body cannot be activated efficiently, and the tumor inhibition effect is poor.

In the pathway enrichment analysis of crossover genes, we found that the m6A crossover genes are highly correlated with m6A typing in the T cell receptor signaling pathway, Yersinia pestis infection, NOD-like receptor signaling pathway, and ubiquitin-mediated protein hydrolysis. These results correspond to our previous study which suggested that T cells were closely involved in the disease progression of pancreatic cancer. We found from the immunohistochemical staining of tissue sections from pancreatic cancer patients that those with stronger RBM15 expression show higher levels of lymphocyte counts (*p* < 0.05). Additionally, we established that patients with stronger expression are more often male and have higher preoperative blood glucose levels (*p* < 0.05). These findings are extremely significant, since they suggest that higher blood glucose levels, which may be related to the need for more energy expenditure to express high pancreatic cancer lesions, promote the elevated expression of RBM15 in pancreatic cancer. A study by Lamkin mentioned that sugar promoted the progression of ovarian cancer [27]. The mechanism may be related to insulin resistance, excessive tumor consumption, etc. Regarding the higher expression of RBM15 in male pancreatic cancer patients, we mentioned previously that the mortality rate of male pancreatic cancer patients is higher than that for women. This suggests that the increased expression of RBM15 is associated with the high mortality rate of male patients. Furthermore, we found that the high RBM15 expression score is not significantly related to the patient’s age of onset, family history, tumor diameter size, neutrophil count, liver and kidney function, and other indicators.

Subsequently, using cellular experiments, we learned that *RBM15* is highly expressed in multiple cell lines. By knocking down the expression of *RBM15* in the CFPAC-1 and BxPC-3 pancreatic cancer cell lines, we found that these two cell lines inhibit cell proliferation, migration, and metastasis. This also suggests that RBM15 promotes the proliferation, migration, and metastasis of pancreatic cancer cells.

## 5. Conclusions

We found that in pancreatic cancer patients with a high expression of RBM15, the disease progression was accompanied by a trend of T lymphocyte aggregation. This was not only through bioinformatics, but also thorough the analysis of clinical data and immunohistochemical staining results of pathological stations. In addition, through a series of in vitro and in vivo experiments, we basically determined the oncogenic role of high expression of RBM15 in pancreatic cancer progression. Therefore, we hypothesize that RBM15 will play an important role in the diagnosis and treatment of pancreatic cancer in the future.

## Figures and Tables

**Figure 1 cancers-15-01084-f001:**
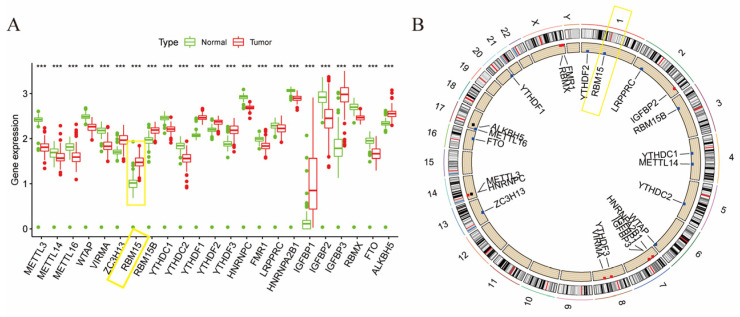
(**A**) Gene expression in normal and cancerous tissue; (**B**) Location of m6A regulators on the chromosome for CNV alteration (*** *p* < 0.001, the yellow frame pointed to the gene *RBM15*).

**Figure 2 cancers-15-01084-f002:**
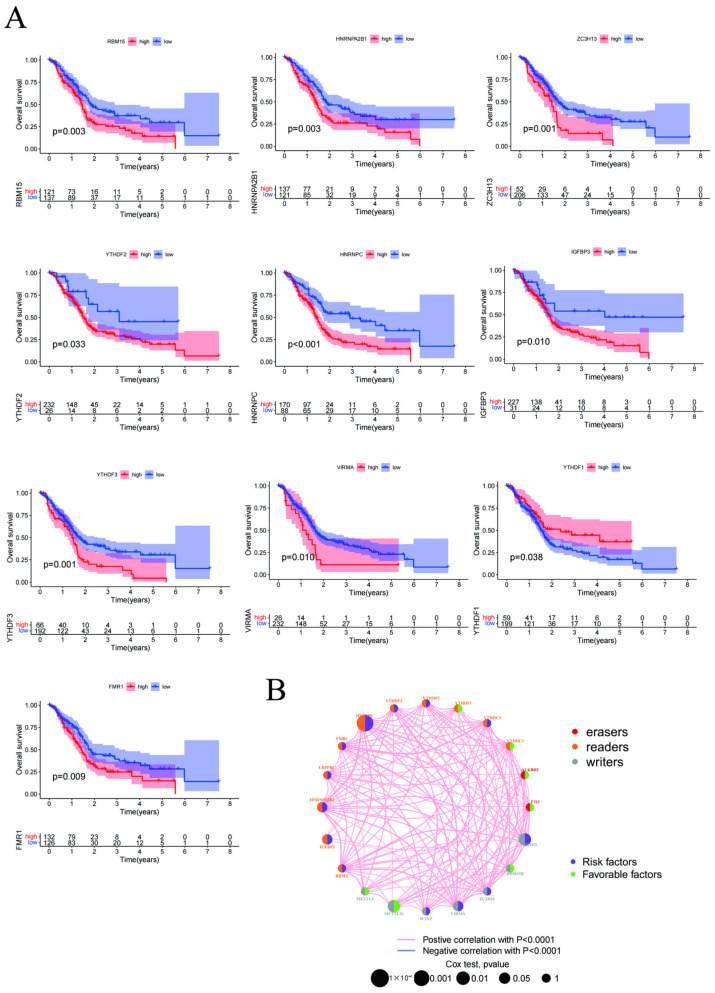
(**A**) Gene expression with survival analysis; (**B**) Analysis of the constructed prognostic network diagram.

**Figure 3 cancers-15-01084-f003:**
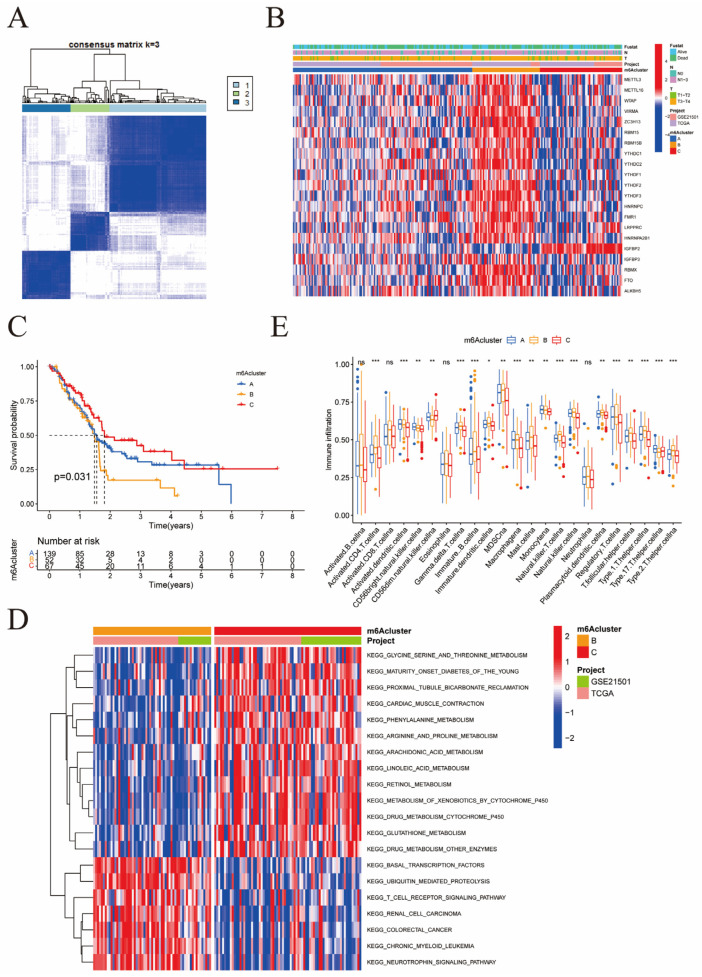
(**A**) Different m6A typings; (**B**) Heat map results made by correlating patients’ clinicopathological features and gene expression; (**C**) Survival analysis of different m6A clusters; (**D**) Biological life information pathways between m6A typing C and B; (**E**) ssGSEA analysis (* *p* < 0.05; ** *p* < 0.01; *** *p* < 0.001).

**Figure 4 cancers-15-01084-f004:**
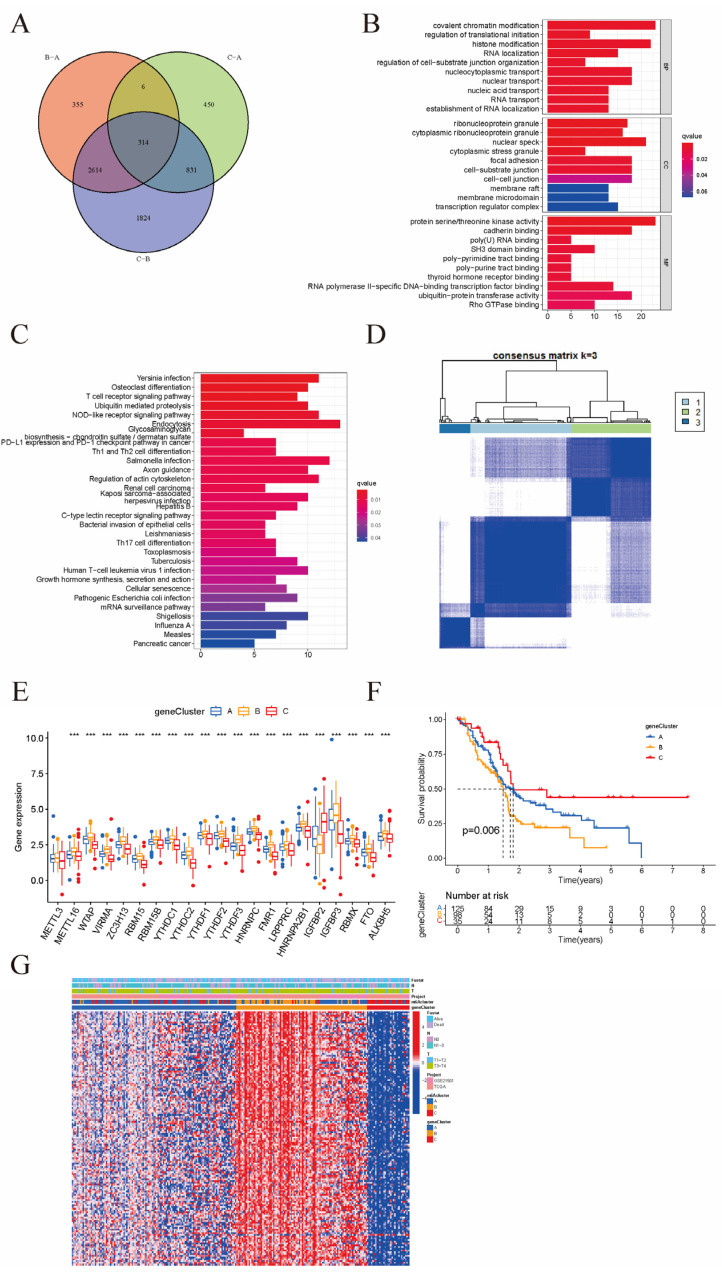
(**A**) Wayne diagram; (**B**) GO enrichment analysis; (**C**) KEGG pathway analysis; (**D**) Prognosis-related gene clusters; (**E**) Expression of m6A-related genes in different gene clusters; (**F**) Survival analysis with different geneCluster; (**G**) Distribution of survival status and disease stage of patients under each type (*** *p* < 0.001).

**Figure 5 cancers-15-01084-f005:**
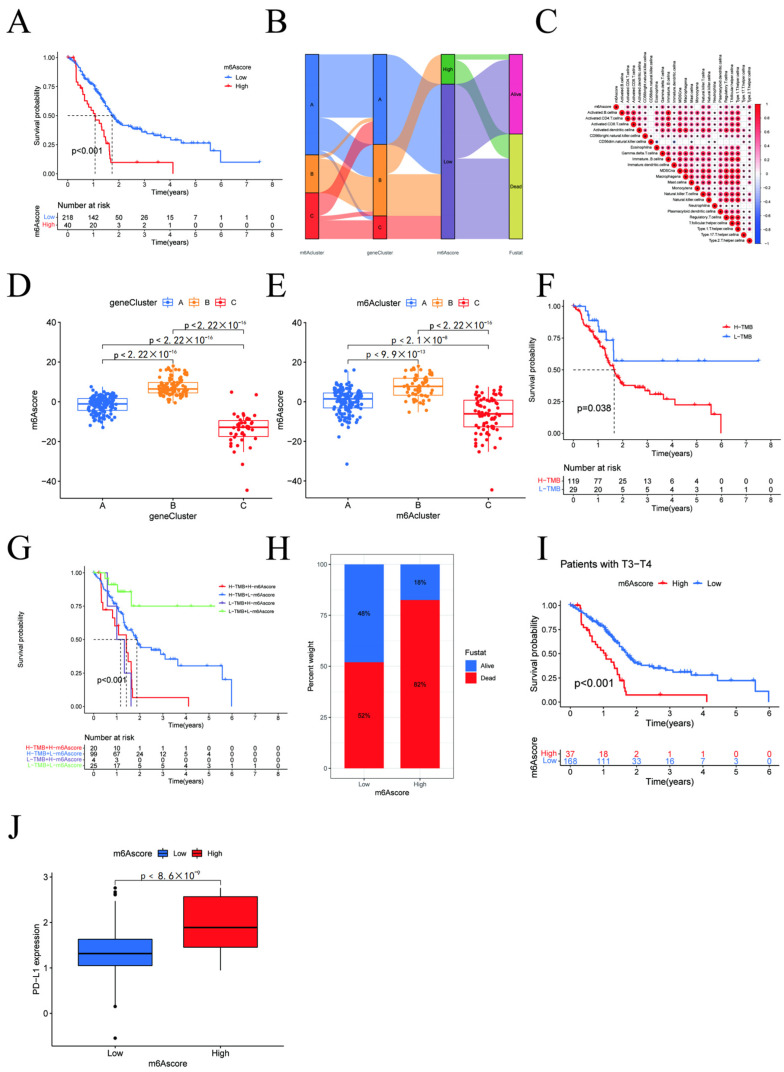
(**A**,**H**) Survival analysis of m6Ascore; (**B**) Alluvial plots; (**C**) Immune correlation analysis; (**D**,**E**) Expression of m6A score for different m6A types; (**F**) Survival analysis for different TMB group; (**G**) Tumor mutation load combined with different m6A score of pancreatic cancer related to survival; (**I**) Survival related to different disease stages; (**J**) PD-L1 gene expression with different m6A scores.

**Figure 6 cancers-15-01084-f006:**
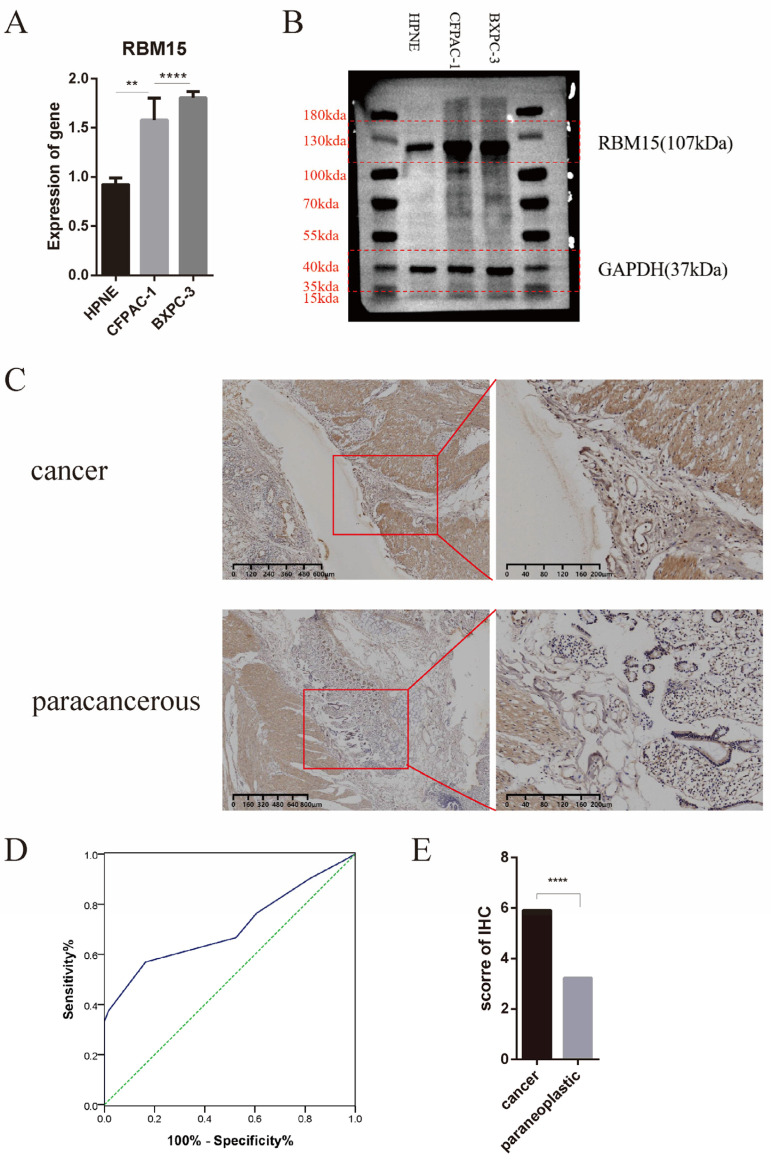
(**A**) Gene expression of RBM15 in cell lines; (**B**) Protein expression of RBM15; (**C**) Immunohistochemical experiments; (**D**) Receiver operating characteristic curve for different immunohistochemical scores; (**E**) Distribution of immunohistochemical scores (** *p* < 0.01; **** *p*< 0.0001). Original blot see Appendix A.

**Figure 7 cancers-15-01084-f007:**
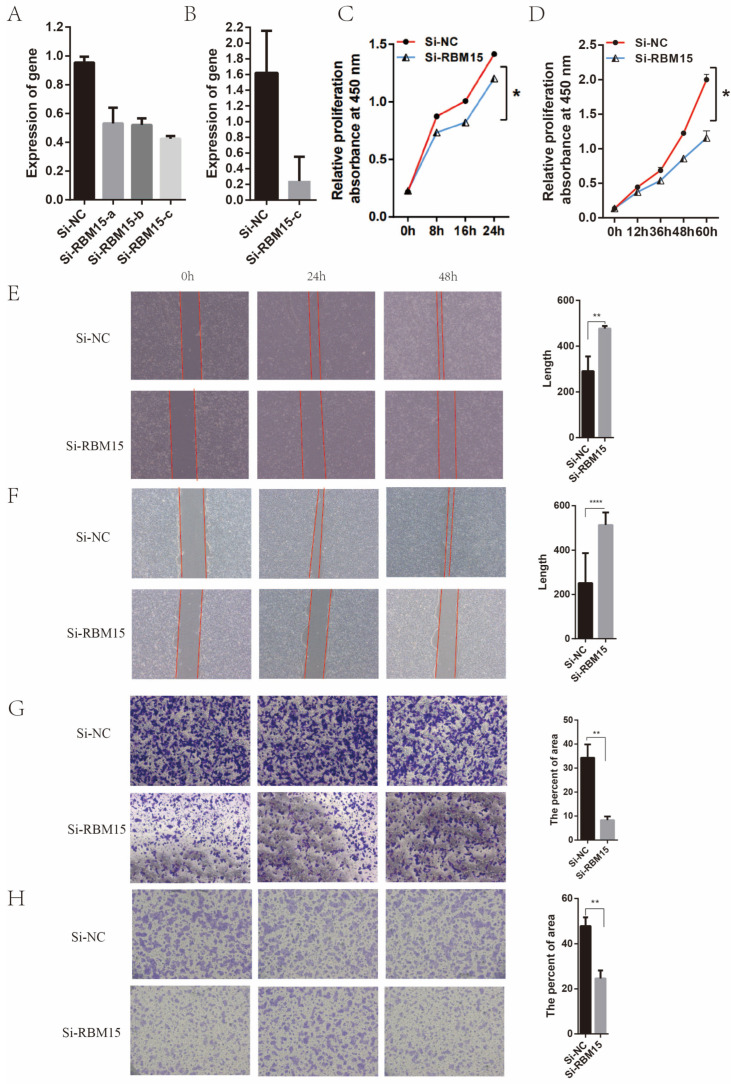
(**A**) Efficiency of different Si-RNAs in knocking down *RBM15* for CFPAC-1; (**B**) Efficiency of Si-RBM15-c in knocking down RBM15 for BxPC-3; (**C**,**D**) Results of CCK-8 experiment in cell lines CFPAC-1 and BxPC-3 after RBM15 gene knockdown; (**E**,**F**) Scratch assay in cell line CFPAC-1 and BxPC-3; (**G**,**H**) Results of Transwell experiments in cell lines CFPAC-1and BxPC-3 (* *p* < 0.05; ** *p* < 0.01; **** *p* < 0.0001).

**Figure 8 cancers-15-01084-f008:**
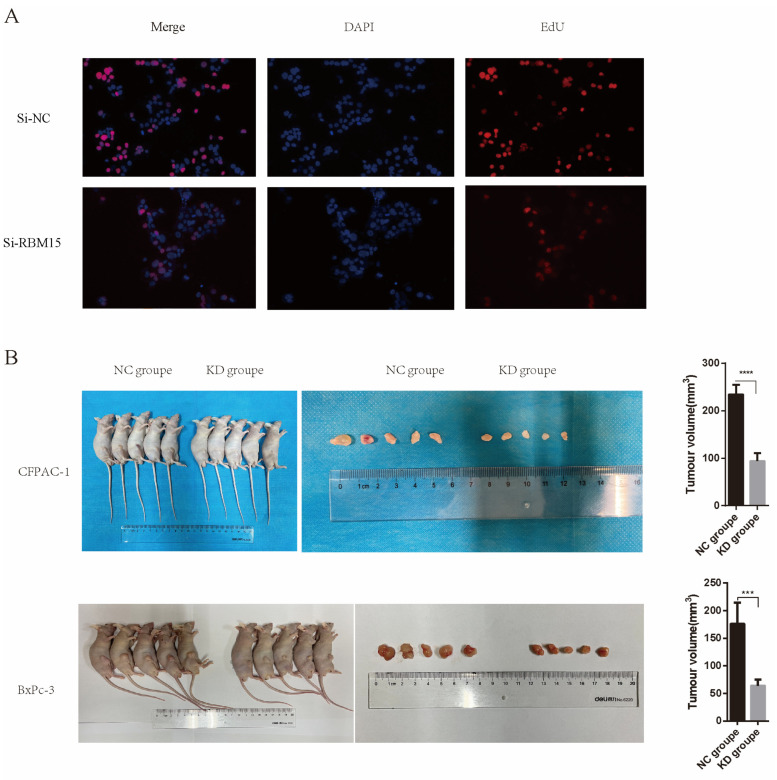
(**A**) EdU test. (**B**) Tumor volumes of different groups in a subcutaneous tumorigenesis model in nude mice (*** *p* < 0.001; **** *p* < 0.0001).

**Table 1 cancers-15-01084-t001:** Relationship between immunohistochemical scores and the clinicopathological characteristics of patients.

		No. of Patients	IHC Score		χ^2^	*p*-Value
			<5	>5		
Gender	Female	30	17 (56.7)	13 (43.3)	3.886	0.049
	Male	42	14 (33.3)	28 (66.7)		
Age	<70 years old	45	20 (44.4)	25 (55.6)	0.094	0.759
	>70 years old	27	11 (40.7)	16 (59.3)		
Smoking history	No	58	23 (39.7)	35 (60.3)	1.407	0.236
	Yes	14	8 (57.1)	6 (42.9)		
History of alcohol consumption	No	62	26 (41.9)	36 (58.1)	0.228	0.633
	Yes	10	5 (50.0)	5 (50.0)		
Family history	None	72	31 (43.1)	41 (56.9)	/	>0.05
Venous blood glucose level	<7 mmol/L	40	22 (55.0)	18 (45.0)	6.1	0.014
	>7 mmol/L	31	8 (25.8)	23 (74.2)		
CA199	<40	14	4 (28.6)	10 (71.4)	0.781	0.377
	>40	53	22 (41.5)	31 (58.5)		
Tumor with or without vascular invasion	No	41	17 (41.5)	24 (58.5)	0.033	0.856
	Yes	28	11 (39.3)	17 (60.7)		
Lymph node metastasis	No	35	10 (28.6)	25 (71.4)	1.82	0.177
	Yes	29	13 (44.8)	16 (55.2)		
Lymphocyte count	1.1–3.2	58	21 (36.2)	37 (63.8)	7.157	0.007
	<1.1	13	10 (76.9)	3 (23.1)		
Tumor size	<3 cm	20	8 (40.0)	12 (60.0)	0.038	0.846
	>3 cm	47	20 (42.6)	27 (57.4)		
Neutrophil ratio	40–75%	62	27 (43.5)	35 (56.5)	0.12	0.73
	>75%	8	4 (50.0)	4 (50.0)		
TNM	0–II	44	17 (29.3)	27 (46.6)	4.59	0.032
	III–IV	14	10 (17.2)	4 (6.9)		

## Data Availability

The data from the GEO database is encoded as GSE21501.

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
