# Peer review of "RBM15 Promates the Proliferation, Migration and Invasion of Pancreatic Cancer Cell Lines"

_cancers, 2023, doi:10.3390/cancers15041084_

Round 1

Reviewer 1 Report (Previous Reviewer 1)

The all content of this manuscript should be carefully revised by some persons with native English language or some professional institution before they submit the revision. 

Author Response

Reviewer 2 Report (New Reviewer)

Specific comments to the authors

In the revised version of the manuscript "RBM15, which Promotes Proliferation, Migration and Invasion of Pancreatic Cancer Cell Lines", the authors were able to address the previously mentioned concerns. Nevertheless, other aspects should/need to be revised before the manuscript can be accepted (see below).

# Title: the corrected title does not reflect the intensive in silico analysis. Please change accordingly.

# Introduction: a clear statement of the proposed objectives of the studies presented should be added at the end of the introduction.

# Material&Methods (page 4 line 152 and 154): Please correct the typos "(The".

# 3.1 Expression of m6A gene in pancreatic cancer: please rewrite which genes are significantly upregulated and which are significantly downregulated to improve readability.

#3.3 Patterns of m6A methylation modifications: After reading this section intensively, it is definitely not clear how types A, B and C are defined by adding a new table. Please clarify by rewording accordingly.

# Figure 6C: What does "paracancerous" mean (like normal tissue). The immunohistochemical stains and images are not good examples as cancerous tissue and nuclear stains of RBM15 are not clear/easy to identify.

# Table 1: Please add more clinicopathological data (e.g. grading, TNM staging, overall survival) and also integrate them into further statistical analysis.

# Discussion: the authors should discuss in more detail why pancreatic cancers with higher TMBs have a worse prognosis than those with low TMBs. The conclusion should focus on their own findings and avoid speculation.

Round 2

Reviewer 1 Report (Previous Reviewer 1)

The current version of the manuscript is acceptable for publication.

Reviewer 2 Report (New Reviewer)

Specific comments to the authors

In the revised version of the manuscript, the authors have endeavoured to address the previously mentioned concerns in a very adequate and convincing manner. The authors have provided additional statistical analyses and detailed explanations to properly address the points of criticism. Therefore, I suggest that the revised manuscript "RBM15, was Identified as a Promoter of Pancreatic Cancer Progression based on Bioinformatics and in Vitro and in Vivo Experiments" should be accepted in the journal "Cancers".

This manuscript is a resubmission of an earlier submission. The following is a list of the peer review reports and author responses from that submission.

Round 1

Reviewer 1 Report

Manuscript entitled ““RBM15, a Prognostic Biomarker and Immunotherapy Target that Correlates With Tumor Immune Microenvironment in Pancreatic Carcinoma” submitted by i Dong H et al. tried to perform the bioinformatic multidimensional analysis using files containing clinical data of patients and m6A-related gene expression differences downloaded from web-based databases, and found that RBM15 plays a key role in the progression of pancreatic cancer by promoting tumor proliferation, migration and metastasis, but there are several flaws in the manuscript.

1.       The linkage between methylation and the immune microenvironment in pancreatic cancer should be explained in the introduction.

2.       In the Materials and Methods, the author should indicate reagent suppliers. What were the search key words used in the manuscript so that 23 m6A-related genes were identified through literature search? Please give us more experiment details of the nude mouse subcutaneous tumorigenesis model including the knockout rbm15 group (Figure 8B) in the Materials and Methods. Also, there should have been more detailed GEO clinical data, 178 or 288 pancreatic cancer patients?

3.       We are also very worried about the Immunoblot image of RBM15 because it is so clean band (Fig. 6B). All figure legends need be rewritten and should improve the quality of all those figures when they were submitted.

4.       Please check English grammar, spelling, and sentence in your manuscript. Finally, all the content of this manuscript should be carefully revised by some persons with native English language or some professional institution before they submit the revision.

Reviewer 2 Report

Authors showed that high expressions of m6A-related genes including RBM15 were poor prognostic factor and related to high expression of PD-L1 mRNA in pancreatic cancer. High expression of intratumoral PD-L1 was a biomarker of PD1/PDL1 inhibitor, but authors' data did not include PD-L1 protein expression. Authors showed the distribution of immune cells according to m6A types. However, m6A-related immune phenotype was not identified. I agree with the relation between tumor aggressiveness and m6A-related gene expressions, but I cannot find the relation between tumor immune microenvironment and m6A-related gene expressions. Poor description for experimental methods should be avoided, but the manner of siRNA transfection and immunohistological analysis were unclear in main text.